## Overview Review

mental health stigma; anti-stigma; intervention; young people; youth mental health; systematic review; meta-analysis

**Corresponding author:**
Ning Song;
Email: umnso@leeds.ac.uk

# The effectiveness of anti-stigma interventions for reducing mental health stigma in young people: A systematic review and meta-analysis

Ning Song[1] , Siobhan Hugh-Jones[2], Robert M. West[1], John Pickavance[2,3] and Ghazala Mir[1]

[1]Faculty of Medicine and Health, Leeds Institute of Health Sciences, University of Leeds, Leeds, UK; [2]Faculty of Medicine and Health, School of Psychology, University of Leeds, Leeds, UK and [3]Centre for Applied Education Research, Bradford Teaching Hospitals NHS Foundation Trust, Bradford, UK

## Abstract

Experiencing mental health stigma during adolescence can exacerbate mental health conditions, reduce quality of life and inhibit young people's help-seeking for their mental health needs. For young people, education and contact have most often been viewed as suitable approaches for stigma reduction. However, evidence on the effectiveness of these anti-stigma interventions has not been consistent. This systematic review evaluated the effectiveness of interventions to reduce mental health stigma among youth aged 10–19 years. The review followed Cochrane and PRISMA guidelines. Eight databases were searched: PubMed, PsycINFO, MEDLINE, Web of Science, Scopus, EMBASE, British Education Index and CNKI. Hand searching from included studies was also conducted. Randomised controlled trials and experimental designs that included randomised allocation to interventions and control groups were included in the review. Narrative synthesis was employed to analyse the results. A meta-analysis was conducted to determine the effectiveness of included interventions. Twenty-two studies were included in the review. Eight studies reported positive effects, 11 studies found mixed effects and 3 studies reported no effect on indicators of mental health stigma among youth. Seven of the effective studies were education-based. Eleven studies were suitable for meta-analysis, and the multivariate meta-analytic model indicated a small, significant effect at post-intervention ($d = .21$, $p < .001$), but not at follow-up ($d = .069$, $p = .347$). Interventions to reduce stigma associated with mental health conditions showed small, short-term effects in young people. Education-based interventions showed relatively more significant effects than other types of interventions.

## Impact statement

We review international evidence on the effectiveness of anti-stigma mental health interventions delivered in schools. Mental health stigma is a global issue, affecting help-seeking, treatment, quality of life and mental health outcomes. In particular, the experience of mental health stigma during adolescence can exacerbate mental health conditions and lead to significant negative life impacts. The evidence base for school-based mental health stigma reduction efforts needs to be widely communicated to researchers, education, healthcare providers and policy makers. Developing evidence in this field about 'what works' will support the growth of governmental policies to ensure anti-stigma interventions reach young people. A focus on anti-stigma interventions in young people has the potential to significantly improve the quality of life of people living with poor mental health as well as their education, employment and help-seeking trajectories. Our article identifies effective components of anti-stigma interventions among youth aged 10–19 years and highlights the lack of evidence from low-and middle-income countries and lack of previous meta-analysis. To the best of our knowledge, ours is the first review to produce an effect size for anti-stigma interventions which target young people. We show that these interventions showed small, short-term effects in young people. Education-based interventions showed relatively more significant effects than other approaches. Our review supports the use of education interventions in schools for reducing mental health stigma in young people and makes recommendations for improving the quality of future interventions and trials. Our article is likely to influence thinking about the adoption of the most appropriate school-delivered prevention strategies for reducing mental health stigma in young people.

## Introduction

Adolescent mental health is a significant global issue. In the global population, of those aged 10–19 years, one in seven have a mental health disorder (WHO, 2021), and 75% of mental health disorders arise before the age of 18 (Patton et al., 2016). However, the proportion of undetected and

untreated cases of mental-ill health is likely to be higher among young people (Kessler et al., 2007). One reason for the low rate of mental health treatment in youth is stigma associated with having a mental health condition (Calear et al., 2021). It has been argued that stigma of any kind is a global phenomenon functioning to 'keep people in the group by enforcing social norms, keep people away from the group as a strategy to avoid disease and keep people down to exploit and dominate' (Hartoga et al., 2020, p. 2). Goffman's (1963) landmark publication described stigma as leading to a 'spoiled identity'. He argued that just as physical marks such as burns or cuts historically identified in slaves or criminals, stigma is a form of social marks to indicate social difference and ostracisation. Contemporary work has begun to focus on culture-specific forms of stigma (of any kind). Yang et al. (2007) argued that everyday life and interactions in a culture indicate 'what matters most' in that context. Termed 'moral experience', the position argues that if a person can engage with 'what matters most' in that context, then they have 'full status' (or personhood) within that cultural group (Kleinman, 2006). Stigma can affect a person's opportunity to participate in 'what matters most', and this may vary by cultural contexts (Yang et al., 2007). What become stigmatised, and the form and practice stigma, will also vary by culture and sub-culture (Yang et al., 2007).

Mental health stigma is defined as 'negative thoughts, beliefs and discriminatory behaviours towards individuals with mental illness or those receiving mental health services' (Pederson et al., 2020, p. 2). 'Public stigma' and 'self-stigma' are the most reported kinds of stigma associated with mental health conditions (Corrigan and Watson, 2002). Societal discrimination or prejudice about mental health conditions has been termed public stigma (Link, 1987), social stigma or enacted stigma (Livingston and Boyed, 2010). Self-stigma, which is also known as internalised stigma (Park et al., 2019), can be defined as the application of negative stereotypes to oneself, resulting in internalised devaluation (Corrigan, 2002).

There is evidence that the experience of mental health stigma during adolescence exacerbates mental health conditions and leads to significant negative life impacts (Yang et al., 2010). Kaushik et al.'s (2016) systematic review on mental health stigma towards children and youth found that when young people hold a viewpoint blaming those with mental-ill health, they are more likely to keep a distance from a young person with a mental health condition. In Moses's (2010) interview study with 56 American youth with mental health conditions, 25 reported experiences of being rejected by peers. American youth reported that they were viewed as lazy by their families when taking medication for a mental health need (Elkington et al., 2012). Nearchou et al. (2018) investigated mental health stigma in Irish youth and found that public stigma predicted lower intention to seek help.

Self-stigma has also been identified as a barrier to help-seeking. For example, Yap et al. (2011) found that Australian youth were less likely to see psychology professionals in school-based mental health services when they considered mental health conditions as an individual weakness. Shechtmana et al. (2018) investigated stigma and help-seeking in Israeli adolescents and found that self-stigma was negatively associated with attitudes towards seeking help. According to a systematic review on psychological outcomes of adolescents' mental health stigma, self-stigma can aggravate a young person's mental health conditions (Ferriea et al., 2020). Additionally, researchers found that, in American adolescents, internalised stigma mediates the relationship between psychosis and subjective quality of life (Akouri-Shan et al., 2022). Mitten et al. (2016) investigated the perceptions of self-stigma in Canadian youth with self-harm experience and reported that youth believed that others would avoid contacting them out of fear of their mental

health conditions. In addition, the main reason for unwillingness to seek help was the worry among American youth about peers' pejorative or stigmatising attitudes towards their help-seeking, as well as concerns with confidentiality in mental health services (Heflinger and Hinshaw, 2016).

The development and evaluation of interventions to challenge mental health stigma in many parts of the world has aimed to increase help-seeking, reduce self-stigma, improve social acceptance and engagement (e.g., in school) and reduce negative impacts of stigma on quality of life and suicide rates. Protest, education and contact have been identified as three main approaches to addressing stigma (Corrigan and Penn, 1999). Protest refers to taking exception to situations where stigmatising experiences occur (Corrigan and Penn, 1999), such as when advocacy and service groups organise events to protest against social stigma. School is a key environment where young people can socialise, obtain knowledge and shape their attitudes and beliefs. Many anti-stigma programmes in Western countries have been mainly designed and delivered in schools. Education interventions provide factual mental health information via teaching and workshops that challenge mental health stereotypes (Morgan et al., 2018). Contact interventions work to lower fear of mental health conditions and develop empathy through the involvement of people who have experience of living with a mental health condition (Pettigrew and Tropp, 2008).

Several theoretical frameworks inform existing interventions; contact-based approaches are informed by Intergroup Contact Theory, coined by Allport (1954), which proposes that intergroup prejudice can be reduced when social groups have more social contact. Pinto-Foltz et al. (2011) stated that their intervention was guided by Fisher's Narrative Paradigm Theory (Fisher and Hood, 1987), which proposes that all meaningful communication is a form of storytelling or reporting of events.

According to a systematic review on anti-stigma interventions with secondary and primary students (Mellor, 2014), education and contact interventions are common choices. Yet outcomes from anti-stigma interventions based on these approaches are inconsistent. For example, Mulfinger et al. (2018) and Pinto-Foltz et al. (2011) evaluated contact-based interventions, respectively, but the former showed positive effects and the later reported no effect. The inconsistent results may link to specific components of the interventions, delivery, dosage and/or characteristics of participants. Studies also often have considerable methodological limitations (Sakellari et al., 2011), meaning caution is needed when interpreting any reports of effectiveness (Mellor, 2014).

There is, therefore, a need to identify the effective components of interventions to reduce mental health stigma in young people. A meta-analysis of high-quality studies would also be valuable in determining the strength of evidence for particular types of interventions. Several systematic reviews have been conducted to investigate the effectiveness of interventions for stigma reduction in young people. However, previous reviews have focused on a specific intervention delivery platform, such as video (Janoušková et al., 2017), have only paid attention to school-based interventions (Mellor, 2014) or have focused on other types of stigma, such as HIV stigma rather than mental health stigma (Hartoga et al., 2020). There are no systematic reviews of mental health anti-stigma interventions conducted via randomised controlled trials for young people or meta-analyses synthesising their effectiveness. This study aimed to fill this evidence gap through a systematic review and meta-analysis to explore the effectiveness of mental health anti-stigma interventions for young people, and where possible, to identify effective interventions with global relevance.

## Methods

This review followed Cochrane and PRISMA guidelines (Page et al., 2021).

### Protocol and registration

A written protocol for the systematic review has been completed and registered on PROSPERO (registration number is CRD42021251932).

### Eligibility criteria and exclusion criteria

The PICOS framework (Amir-Behghadami and Janati, 2020) was employed to identify eligibility and exclusion criteria. Table 1 demonstrates the selection criteria.

### Information sources

Eight databases were searched to identify eligible studies: PubMed, PsycINFO (2002 to present), MEDLINE (1950 to present), Web of Science (1999 to present), Scopus (1823 to present), EMBASE (1996 to present), British Education Index (1975 to present) and the Chinese database CNKI (1999 to present). A Chinese database was included to address the lack of evidence from China in existing reviews because of the availability of language expertise within the research team. A manual search was conducted to identify further eligible studies by examining reference lists of included papers. Any potential papers were found via this search and then screened following the procedures described below.

The search was conducted from May 2021 to July 2021, and a manual trace back literature search was conducted in March 2022. No limits were applied to publication year or language for the database search. However, if studies were not written in English or Chinese, they were excluded from the review.

### Search strategy

Search terms were determined around four domains: stigma, mental health conditions, young people and intervention. Appendix 1 of the Supplementary Material specifies the search terms.

### Study selection

This review followed the PRISMA guidance (Page et al., 2021) for study selection. All documents and data from reviewed papers were stored in Rayyan (Ouzzani et al., 2016). Following search returns, duplicate papers were removed, and the titles and abstracts of the remaining papers were screened. Double screening was also conducted. Eligible papers were then subjected to full-text review. Papers without full access were sought by contacting authors for copies. The full-text papers were screened according to the inclusion criteria. The second reviewer reviewed a random 50% of the full-text papers. Disagreements were resolved through discussion.

### Data extraction

Data were extracted using the Cochrane Collaboration-recommended templates 'Data Collection for Intervention Reviews for Randomised Controlled Trials Only' (Cochrane, 2021). Tables 2 and 3 provide information on study characteristics of included studies.

### Quality assessment

Given that the included studies were randomised controlled trials, cluster-randomised trials and quasi-experimental designs involving intervention and control groups with random allocation of participants, Risk of Bias 2 (Sterne et al., 2019) was employed for the quality assessment of randomised controlled trials and quasi-experimental designs (Cochrane, 2021), and Risk of Bias 2 Cluster-Randomised Trials (Sterne et al., 2019) was used for the quality of assessment of cluster-randomised trials. After assessing the quality of included studies, the second assessor conducted a double quality assessment. Disagreements were resolved by discussion.

### Data synthesis

A narrative synthesis was used to analyse the findings of the eligible studies and the components of the reported interventions. This allowed results of included studies to be assessed systematically and comprehensively and significant features of the included studies to be highlighted (Ryan, 2013).

Meta-analysis was performed using *R* (v4.2.0). Models were fit using the *metafor* (v3.4.0) package with covariance imputed for robust estimation using *clubSandwich* (v0.5.6). The sample size, mean and standard deviation were obtained for control and intervention groups at up to three time points for all screened studies. They were *Pre-Intervention*, *Post-Intervention* and *Follow-Up*. All measures were transformed such that a positive mean difference indicated a reduction in stigma. Cohen's *d* was then computed as the standardised mean difference at each time point available for each study (Cohen, 1988). Firstly, a multivariate meta-analysis model on *Pre-Intervention* effect sizes was conducted as a control. In many studies, multiple measures of stigma were taken (see Tables 2 and 3). Therefore, a random effect was specified to better account for the variance both within and between the studies

**Table 1.** Eligibility and exclusion criteria

|  | Eligibility | Exclusion |
|---|---|---|
| Population | Mean age between 10 and 19 years old with any gender and ethnicity. | Mean age younger than 10 years old or older than 19 years old. |
| Intervention | Any interventions/programmes/campaigns for mental health stigma reduction. | Anti-stigma interventions/programmes/campaigns not related to mental health, such as anti-stigma for HIV or disability. |
| Comparator | At least one control group. | No comparator. |
| Outcome | The degree of reduction of mental health stigma. | Degree of stigma reduction was not related to mental health. |
| Study design | Randomised controlled trials, cluster-randomised trials and quasi-experimental designs. | Other study designs, such as qualitative study and case study. |

**Table 2.** Characteristics of included studies

| Study citation | Study design | Intervention(s) | N Age (years) | Delivery | Stigma-related measure | Post-test effect size (Cohen's *d*)/*p*-value | Follow-up effect size (Cohen's *d*)/*p*-value |
|---|---|---|---|---|---|---|---|
| Cangas et al. (2017) | QED | Education | 552 14–18 | Video game | Questionnaire on Student Attitudes towards Schizophrenia | *p* = .000 (Dangerousness) *p* = .001 (Stereotypes) | NA |
| Perry et al. (2014) | CRT | Education | 380 Mean 14.94 | Programme teachers | The Depression Stigma Scale | Interaction effect: *p* < .05 | Interaction effect: *p* < .05 |
| Nguyen et al. (2020) | RCT | Education | 3,000 Mean 15 | Trained teachers | Mental Health Knowledge and Attitude Test | *p* < .0001 (Vietnamese) *p* < .0001 (Cambodian) | NA |
| Link et al. (2020) | CRT | Education | 416 NI | Teachers | Knowledge and Positive Attitudes Children's social distance | *p* < .001 (Attitudes) *p* < .05 (Social distance) | *p* < .001 (Attitudes) *p* < .05 (Social distance) |
| Winkler et al. (2017) | RCT | Education Contact | 499 Mean 18.41 | A mental health professional (psychiatrist or case manager) and an expert by experience | The Community Attitudes towards Mental Illness Reported and Intended Behaviour Scale | Seminar arm: *d* = .61 (Attitudes); *d* = .58 (Behaviour) Video arm: *d* = .49 (Attitudes); *d* = .26 (Behaviour) | Seminar arm: *d* = .43 (Attitudes); *d* = .26 (Behaviour) Video arm: *d* = .22 (Attitudes); *d* = .21 (Behaviour) |
| Mulfinger et al. (2018) | RCT | Contact | 98 13–18 | A young adult peer with mental illness and a young mental health professional | Stigma Stress Scale | *p* < .001 | *p* < .001 |
| Economou et al. (2014) | RCT | Education | 1,081 13–15 | Two psychologists, trained in child psychology and group dynamics | Alberta Pilot Site Questionnaire Toolkit; Social Distance measure | *p* < .001(Attitudes) *p* < .001 (Social distance) | NA |
| Milin et al. (2016) | RCT | Education | 534 NI | Trained teachers | Attitudes towards Mental Illness | Interaction effect: *p* < .01 | NA |
| Staniland and Byrne (2013) | QED | Education + Contact | 395 NI | The author | Adjective Checklist; Shared Activities Questionnaire | *p* < .001 (Attitudes) NI (Behaviour) | *p* = .01 (Attitudes) *p* = .22 (Behaviour) |
| Vila-Badia et al. (2016) | RCT | Contact | 280 14–18 | Healthcare staff | The Community Attitudes towards Mental Illness | *p* = .000 (Authoritarianism) *p* = .742 (Benevolence) *p* = .019 (Social restrictiveness) *p* = .117 (Community mental health ideology) | NA |
| O'Mara et al. (2013) | RCT | Education | 294 NI | Trained peer facilitators | The Stigma Scale – Attribution Questionnaire Revised | *p* < .001 (Low-need schools) *p* > .05 (Overall schools) | NA |
| Gonçalves et al. (2015) | RCT | Education | 207 NI | Treatment group | Self-Stigma of Seeking Help Scale; Social Stigma for Receiving Psychological Help Scale; Attribution Questionnaire-Children form | *p* < .05 (Self-stigma) *p* < .05 (Social stigma) *p* < .05 (Attribution) | *p* > .05(Self-stigma) *p* > .05 (Social stigma) *p* > .05 (Attribution) |
| Economou et al. (2012) | RCT | Education | 616 13–15 | An educational psychologist and a psychiatrist, especially trained in group dynamics | Alberta Pilot Site Questionnaire Toolkit; Social Distance measure | *p* < .05 (Attitudes) *p* < .05 (Social distance) | *p* < .05 (Attitudes) *p* > .05 (Social distance) |

**Table 2.** (*Continued*)

| Study citation | Study design | Intervention(s) | N Age (years) | Delivery | Stigma-related measure | Post-test effect size (Cohen's *d*)/*p*-value | Follow-up effect size (Cohen's *d*)/*p*-value |
|---|---|---|---|---|---|---|---|
| DeLuca (2020) | CRT | Education + Contact | 232 13–18 | NI | The Perceptions of Stigmatisation by Others for Seeking Help scale; The Self-Stigma of Seeking Help scale | NI | Anticipated stigma: *p* = .020 Self-stigma: *p* > .05 |
| Chisholm et al. (2016) | CRT | Education | 769 12–13 | NI | The Reported and Intended Behaviour Scale | *p* = .5 | *p* = .03 |
| Ahmad et al. (2020) | CRT | Education | 731 Mean 17.4 | NI | The Attitude scale; The Social Distance scale; The Positive Action scale | *p* = .010 (Attitudes) *p* > .05 (Social distance) *p* < .001 (Action) | NA |
| Painter et al. (2017) | QED | Education; Contact; Printed material | 721 Mean 11.5 | Teachers; Two college students with a history of bipolar I disorder and bipolar II disorder. | NI | Printed materials: *p* > .05 NI | NA |
| Cheetham et al. (2020) | RCT | Education | 463 Mean 14.94 | NI | A five-point stigma scale | NI | *p* = .171 (Weak no sick) *p* = .242 (Dangerousness) Interaction effect: *p* < .001 (Weak no sick) |
| Saporito et al. (2011) | QED | Education + Contact | 159 Mean 14.76 | Ten trained graduate and undergraduate psychology students | Community Attitudes towards the Mentally Ill; Attitudes towards Seeking Professional Psychological Help; Implicit Association Test | *p* = .03 (Attitudes to mental health) *p* = .001 (Attitudes to treatment) *p* > .05 (Implicit Attitudes to mental health) *p* > .05 (Implicit Attitudes to treatment) | NA |
| Townsend et al. (2019) | RCT | Education | 6,025 NI | High school teachers | Reported and Intended Behaviours Scale | NI | *p* = .08 |
| Pinto-Foltz et al. (2011) | CRT | Contact | 156 13–17 | Trained consumers who were recovery from mental illness | A five-item subscale of the Revised Attribution Questionnaire | *p* = .33 | *p* > .05 |
| Howard et al. (2018) | RCT | Education | 327 16–19 | NI | Self-Stigma for Depression Scale | *p* > .05 | NA |

Note: CRT, cluster-randomised trial, NA, not available, NI, no information, QED, quasi-experimental design, RCT, randomised controlled trial.

**Table 3.** Outcomes and intervention sessions

| Study citation | Country | Target | Primary or only outcome | Other outcomes (no primary outcome indicated) | Intervention sessions | Indicators of intervention compliance | Comparator intervention(s) | Control condition |
|---|---|---|---|---|---|---|---|---|
| Cangas et al. (2017) | Spain | Schizophrenia | Stigmatising attitudes towards schizophrenia. | NI | 12 sessions × 60 min | NI | NA | Another video game unrelated to mental health |
| Perry et al. (2014) | Australia | Depression | Mental health literacy | Stigma, help-seeking, psychological distress and suicidal ideation | 10 h | Data were obtained from 380 participants at baseline, 322 participants post-intervention and 208 participants at a 6-month follow-up | NA | No intervention control condition |
| Nguyen et al. (2020) | Vietnam and Cambodia | Mental health | NI | Knowledge and stigma about mental health | NI | 89% of students in intervention group and 78% in control group provided data | NA | No intervention control condition |
| Link et al. (2020) | USA | Mental health | NI | Knowledge and attitudes towards menial health and social distance | NI | 75% participants completed assessment at 24 months | Contact; Printed materials | No intervention control condition |
| Winkler et al. (2017) | Czech Republic | Mental health | NI | Stigma-related attitudes and behaviours | 1 session | 68.4% in seminar and 73.1% in video completed assessment | NA | Active control group-received leaflet |
| Mulfinger et al. (2018) | German | Mental health | Stigma stress; quality of life | Empowerment; self-stigma, disclosure-related distress, empowerment, help-seeking intentions, recovery and depressive. | 3 sessions × 2 h | 86% completed post-assessment and 78% completed follow-up assessment | NA | Treatment as usual |
| Economou et al. (2014) | Greece | Schizophrenia | NI | Adolescents' beliefs, attitudes and desired social distance | 1 session × 120 min | NI | NA | Received a 2-h discussion on immunisation |
| Milin et al. (2016) | Canada | Mental health | NI | Mental health knowledge and attitudes towards mental illness/ stigma | NI | 87.8% completed both pre- and post-questionnaires | NA | Teaching as usual |
| Staniland and Byrne (2013) | Australia | Autism | NI | Autism knowledge, attitudes towards disabilities, behavioural intentions | Six sessions × 50 min | NI | NA | No-intervention non-peer |
| Vila-Badia et al. (2016) | Spain | Mental health | Social stigma towards mental health | NI | One session | NI | NA | No intervention control condition |
| O'Mara et al. (2013) | Canada | Mental health | NI | Stigma and depression | 1 session × 75 min | 91.2% overall completion rate | NA | NI |
| Gonçalves et al. (2015) | Portugal | Mental health | NI | Self-stigma, social stigma, attribution for mental health | 1 session | NI | NA | NI |
| Economou et al. (2012) | Greece | Schizophrenia | NI | Participants' beliefs and attitudes; social distance | 1 session | NI | NA | A talk about nutrition and healthy living |
| DeLuca (2020) | USA | Mental health | Negative stereotypes; intended social distance; knowledge; | Anticipated stigma; self-stigma; disclosure worries | 1 session | 89% of students participated in the study | NA | Received a presentation of parallel length on |

(Continued)

**Table 3.** (*Continued*)

| Study citation | Country | Target | Primary or only outcome | Other outcomes (no primary outcome indicated) | Intervention sessions | Indicators of intervention compliance | Comparator intervention(s) | Control condition |
|---|---|---|---|---|---|---|---|---|
| | | | negative effects; help-seeking intentions | | | | | 'careers in psychology' |
| Chisholm et al. (2016) | UK | Mental health | Stigma of mental illness | Knowledge of mental illness; emotional wel-lbeing; resilience; help-seeking; acceptability | 1session (1 day) | 14.6% dropout | Education + Contact | NI |
| Ahmad et al. (2020) | USA | Mental health | NI | Knowledge; attitudes; social distance; positive actions | Weekly/biweekly | 58.9% provided data at T1 and one additional time point (T2 or T3) | In delayed condition | NA |
| Painter et al. (2017) | USA | Mental health | NI | Stigmatising attitudes, beliefs, behaviours and behavioural intentions and recognition of mental illnesses and favourable attitudes towards help-seeking | Curriculum (3–6 days per period) | NI | Education; Contact; Printed material | No intervention control condition |
| Cheetham et al. (2020) | Australia | Alcohol misuse | | Stigma; help-seeking; confidence; alcohol use | 1 session | NI | NA | NI |
| Saporito et al. (2011) | USA | Mental health | NI | Community Attitudes towards the Mentally Ill; Attitudes towards Seeking Professional Psychological Help; Implicit Bias; Semantic Differential Willingness to Seek Treatment; Treatment Information; Positive and Negative Affect | 1 session × 35 min | NI | NA | A parallel 35-min educational presentation with content unrelated to mental health |
| Townsend et al. (2019) | USA | Depression | NI | Depression knowledge, mental health stigma | NI | NI | NA | NI |
| Pinto-Foltz et al. (2011) | USA | Mental health | NI | Mental health stigma and literacy | NI | 8% of participants failed to complete the standard measures at all time points | NA | No intervention control condition |
| Howard et al. (2018) | Australia | Depression | Anticipated self-stigma for depression | Help-seeking intentions; Causal attribution; Depressive symptoms | NI | 93.2% completed the intervention. | NA | Neutral information on the symptoms of depression |

Note: NA, not available; NI, no information.

(Harrer et al., 2021). Moreover, robust variance estimation was used to account for the fact that measures within the studies were likely to be highly correlated having been generated from the same sample or intervention. The correlation coefficient was set to .6 a priori, but subsequent fits with coefficients ranging from .2 to .8 showed no discernible difference. The variance of the distribution of effect sizes ($\tau^2$) was calculated using restricted maximum likelihood. Once it was confirmed that there were no effects at baseline, separate multivariate meta-analysis models were fit to estimate pooled effect sizes at *Post-Intervention* and *Follow-Up* using the same specifications as above. Finally, for completeness, *Post-Intervention* and *Follow-Up* effects were combined, and a multivariate meta-analysis model was fit with time as a moderating effect. This was coded as a dummy variable with *Post-Intervention* as the intercept and *Follow-Up* as the coefficient. This was to determine whether any decline in the efficacy of interventions was significant.

## Results

### Study selection

A PRISMA flow diagram (see Figure 1) shows the procedures for study selection. In total, 170 studies were identified for full-text screening. Of these, 21 studies were excluded as full texts were not accessible, leaving 149 for full-text reviews. The agreement between two reviewers for study selection was calculated at a full-text review using Cohen's *k* coefficient, and the value was 0.9, suggesting excellent interrater reliability. Following this, 16 studies were assessed as eligible for the review. Other reasons for exclusion included studies that were protocols or not relevant to the review topic. A further six studies were identified via a hand searching from the included studies. A final total of 22 publications were included in the review.

### Study characteristics

#### Population characteristics

The interventions were conducted in 14 countries and mainly distributed in Europe (*n* = 8), North America (*n* = 9), Australia (*n* = 4) and Asia (*n* = 1). The number of participants including control groups ranged from 98 to 6,025. The age range of study samples was 12–19 years. Seven studies indicated the mean age of participants ranged from 14.52 to 18.41, and six studies did not report participants' age but stated that these were secondary/high school students. Most studies investigated the efficacy of interventions for both boys and girls with two exceptions: Staniland and Byrne (2013) only included boys and Pinto-Foltz et al. (2011) recruited only girls. Except for Mulfinger et al. (2018), who

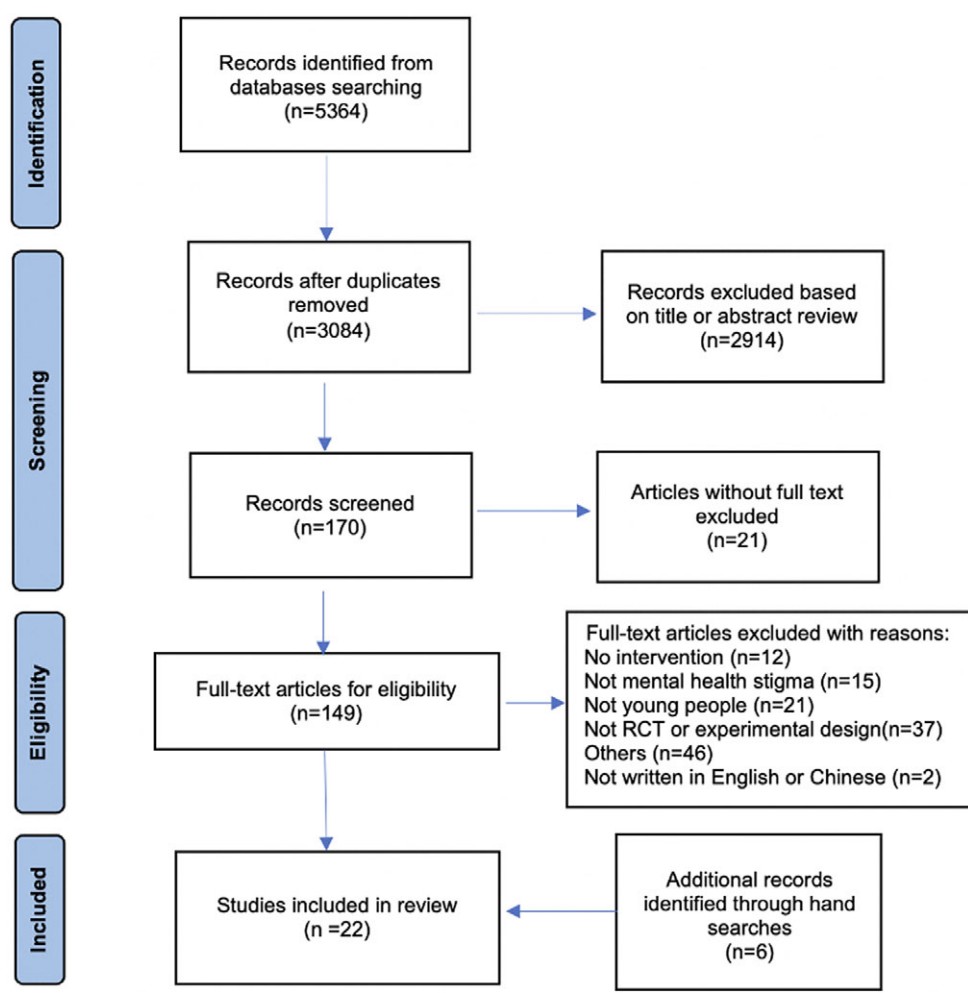

**Figure 1.** PRISMA flow diagram.

recruited participants with a diagnosed mental health condition, all other study samples were community groups. Table 2 shows the overall characteristics of included studies.

### Intervention characteristics

Seven interventions aimed to reduce stigma linked to a specific mental health condition (such as depression); one targeted stigma about autism and the remaining interventions aimed to reduce stigma around general mental health conditions. One study (Mulfinger et al., 2018) was conducted in an inpatient setting, and the other 21 studies were implemented in secondary/high schools.

Eighteen studies used randomised controlled trials, including cluster-randomised trials to evaluate intervention efficacy, and the other four interventions used experiment designs involved random allocation of participants to intervention and control groups but did not use the term randomised controlled trials.

Four studies had comparator intervention(s) (Chisholm et al., 2016; Painter et al., 2017; Ahmad et al., 2020; Link et al., 2020). Participants were grouped in no intervention control condition in seven studies (Pinto-Foltz et al., 2011; Staniland and Byrne, 2013; Perry et al., 2014; Vila-Badia et al., 2016; Painter et al., 2017; Link et al., 2020; Nguyen et al., 2020). Eight studies adopted active control groups, where participants received a leaflet (Winkler et al., 2017), a talk about nutrition and healthy living (Economou et al., 2012), a presentation of parallel length on 'careers in psychology' (DeLuca, 2020), treatment (not described) as usual (Mulfinger et al., 2018), a parallel 35-min educational presentation with content unrelated to mental health (Saporito et al., 2011), a 2-h discussion on immunisation (Economou et al., 2014), neutral information on the symptoms of depression (Howard et al., 2018) and another video game unrelated to mental health (Cangas et al., 2017).

Thirteen studies reported some details on intervention compliance. Table 3 shows information on outcomes, intervention sessions, post-intervention time, indicators of intervention compliance, comparator interventions and control condition.

Included interventions were mainly based on education, contact or education plus contact. Education interventions provided factual mental health information via teaching and workshops that aimed to challenge mental health stereotypes (Morgan et al., 2018). Contact interventions aimed to lower fear and develop empathy through contact with people living with mental health conditions (Pettigrew and Tropp, 2008). Sixteen interventions were education-based, and they were delivered by trained teachers/professionals/facilitators via the curriculum, lessons, discussions, lectures, activities, meetings and seminars. The delivery forms were interactive, which meant participants needed to interact with facilitators/peers, such as participating in a discussion, rather than passively receiving content from facilitators. Five studies were contact-related interventions, inviting people who had poor mental health to share their lived experience or personal story via video delivery or presentation (Pinto-Foltz et al., 2011; Vila-Badia et al., 2016; Painter et al., 2017; Winkler et al., 2017; Mulfinger et al., 2018). Three interventions included both education and contact elements (Saporito et al., 2011; Staniland and Byrne, 2013; DeLuca, 2020).

Interventions were commonly delivered via an educational curriculum approach; this introduced symptoms and basic information on mental health conditions to increase mental health knowledge, interaction and discussion on how to reduce stigma and videos that shared the experience of people living with mental conditions. Cangas et al. (2017) delivered the intervention using a video game that featured characters with various mental health conditions (e.g., schizophrenia and depression), presenting knowledge on mental health conditions to correct participants' misunderstanding or stereotypes. Guidebooks, leaflets, booklets, printed materials and some supplementary resources on stigma reduction were also incorporated into education, contact or education plus contact interventions. Participants could access these materials online through a website designed for the intervention. Three studies (Staniland and Byrne, 2013; Painter et al., 2017; Link et al., 2020) also introduced homework exercises.

Fifteen studies reported the intervention delivery agent (see Table 2). Teachers who were trained to deliver the intervention were reported in six studies (Perry et al., 2014; Milin et al., 2016; Painter et al., 2017; Townsend et al., 2019; Link et al., 2020; Nguyen et al., 2020). Four studies showed that their delivery agents were mental health professionals, psychologists or psychiatrists (Economou et al., 2012; Economou et al., 2014; Winkler et al., 2017; Mulfinger et al., 2018). Other delivery agents included the researcher (Staniland and Byrne, 2013), trained peer facilitator (O'Mara et al., 2013) and trained graduate and undergraduate psychology students (Saporito et al., 2011). Contact elements were delivered by healthcare staff (Vila-Badia et al., 2016), and young people and adults who had a mental health condition (Pinto-Foltz et al., 2011; Painter et al., 2017; Mulfinger et al., 2018).

Thirteen studies reported how many sessions the intervention involved. Ten interventions were single session only. The longest intervention was $12 \times 60$ min sessions (Cangas et al., 2017). Among those who reported intervention dose ($k = 17$), the shortest and longest interventions lasted 2 min (Winkler et al., 2017) and 120 min (Economou et al., 2014), respectively.

All studies collected post-intervention data. Nine studies collected data immediately after the intervention was completed. Other studies assessed their outcomes at different time points. Twelve studies collected follow-up data. Pinto-Foltz et al. (2011) and DeLuca (2020) had two follow-up points, and Cheetham et al. (2020) had three. The longest follow-up point was 24 months (Link et al., 2020), and the shortest one was 3 weeks (Mulfinger et al., 2018).

The intervention outcomes examined in the studies included: self-stigma, stigmatising beliefs/attitudes, stigma stress, mental health knowledge, help-seeking intentions, social distance, disclosure worries, suicidal ideations, alcohol use, resilience, quality of life, acceptability, confidence, implicit bias, empowerment and recovery. Some studies investigated outcomes related to specific mental health conditions, such as stigmatising attitudes towards schizophrenia.

Most studies did not report a primary outcome or distinguish between the primary and secondary outcomes; only five studies distinguished and reported these. Two of these studies investigated stigma-related outcomes as the primary outcomes, whereas these were secondary outcomes in the other two studies. Researchers used different standardised measures to measure outcomes. Our review focused on stigma-related outcomes for which results are reported in Table 3.

### Intervention effectiveness

In total, eight interventions showed positive effects, including six education-only intervention, one contact-only intervention and one intervention that included education and contact. Six education-based interventions reported significant stigma reduction at immediately after the intervention (Perry et al., 2014; Milin et al., 2016; Cangas et al., 2017; Winkler et al., 2017; Nguyen et al.,

2020), and 2 weeks afterwards (Economou et al., 2014). Of these, the following five interventions were conducted in classrooms and aimed to both deliver mental health literacy and correct misconceptions about mental health conditions. Perry et al. (2014) delivered an anti-stigma intervention to Australian youth and found significant effects on stigma reduction at different test time points ($p < .05$) compared with the control group. Intervention delivery was via a booklet, slideshow and various appendices in class. Nguyen et al. (2020) evaluated an intervention for Vietnamese and Cambodian youth and reported significant effects on stigma reduction in both Vietnamese ($p < .0001$) and Cambodian youth ($p < .0001$) at post-intervention compared to the control group. This intervention involved six modules that were on mental health-related knowledge and responses to mental health conditions. Link et al. (2020) evaluated the efficacy of a curriculum intervention compared with two comparator interventions (contact and printed materials) and a control group. The curriculum intervention involved a didactic component group discussion and homework exercises in each module. The curriculum intervention significantly increased knowledge and improved attitudes towards mental health conditions ($p < .001$) and reduced social distance ($p < .05$), with these effects being maintained over 2 years. Only the curriculum aspect of the intervention was effective, and neither contact nor printed materials showed significant effects on measured outcomes. Economou et al. (2014) obtained significant reductions in the intervention group in negative attitudes towards schizophrenia at post-intervention ($p < .001$) and social distance ($p < .001$). This intervention was delivered through an educational talk in class. Milin et al. (2016) intervention for stigma reduction in youth also obtained positive findings showing an increase of positive attitudes towards mental health conditions compared to the control group ($p < .01$). Researchers delivered this intervention via six mental health stigma-related modules, which were embedded in classroom activities.

Winkler et al. (2017) evaluated two interventions (seminar and short video) and an active control group (leaflets). At post-intervention, there were small effects in the flyer arm, medium in the seminar arm and medium in the video arm. At a 3-month follow-up, there were medium effects in the seminar arm and small effects in the video arm but no effect in the flyer arm. The seminar showed the strongest and relatively most stable effect on outcomes, which suggested that the role of facilitators could be of importance in changing attitudes. One study (Cangas et al., 2017) designed a video game to deliver the education intervention. They reported a statistically significant stigma reduction towards schizophrenia (dangerousness: $p = .000$; stereotypes: $p = .001$). Finally, Mulfinger et al. (2018) conducted a contact-based intervention in an inpatient setting. This peer-led programme covered five themes aiming to increase disclosure of mental health conditions. The intervention was delivered by a young adult with experience of a mental health condition and a young mental health professional. At post-intervention, there was a significant improvement in stigma stress (i.e., person feels stigma-related harms outweigh the coping resources; the level of stigma stress will be high if that person feels less confident to cope with stigma; $p < .001$) as well as at a 3-week follow-up ($p < .001$).

Eleven interventions reported mixed effects. Of those, the following six studies evaluated education-based interventions. O'Mara et al.'s (2013) intervention involved videos followed by researcher-facilitated discussion and focus groups. It did not significantly reduce stigma overall but showed significant decrease stigma in the low-needs schools (i.e., less needy in the Learning

Opportunity Index), resulting in lower rates of depression and more uses of healthy coping strategies after the intervention ($p < .001$). Gonçalves et al. (2015) reported significantly higher intervention effects on self-stigma ($p < .05$), social stigma ($p < .05$) and attribution ($p < .05$) for the intervention group than the control group, but at a 1-month follow-up, the effects diminished for these three outcomes to become non-significant. Economou et al.'s (2012) intervention involved discussions delivered by an educational psychologist and a psychiatrist. They found a significant change in participants' beliefs and attitudes towards people with schizophrenia ($p < .05$), and this effect was retained at a 12-month follow-up ($p < .05$). The effect on social distance ($p < .05$) was shown at post-intervention but not at the follow-up point.

Ahmad et al. (2020) evaluated a school club where students engaged in club activities and meetings and reported that the intervention had overall effects on attitudes towards mental health conditions (intervention group: $p = .010$; delayed group [i.e., received the intervention later]: $p = .004$), but from the second to third test time point, the effects were non-significant. A significant overall improvement in positive actions was also found (intervention group: $p < .001$; delayed group: $p < .001$). For social distance outcome, significant effects over time were found in the delayed group ($p = .037$) but did not in the intervention group. Cheetham et al. (2020) reported an intervention providing information on mental health condition and help-seeking did not demonstrate its efficacy over 12 months in reducing stigmatising attitudes towards mental health for both 'weak not sick' ($p = .171$) and 'dangerousness' ($p = .242$). However, compared with the control group, more stigma reduction was reported ($p < .001$). One study (Vila-Badia et al., 2016) was contact-only content. They found that the intervention had positive effects on authoritarianism (i.e., a viewpoint that people who are living with mental health conditions are inferior; $p = .000$) and social restrictiveness (i.e., the attitude that individuals with mental health conditions are a danger to society and should be restricted during or after hospitalisation; $p = .019$), two factors in *The Community Attitudes towards Mental Illness* (Taylor and Dear, 1981), but had no effect on benevolence (i.e., attitudes that include encouragement and paternalism towards people living mental health conditions; $p = .742$) and community mental health ideology (i.e., beliefs that people with mental health conditions should integrate into society in general; $p = .117$). Participants in this intervention were shown a documentary film that was related to mental health conditions. Three studies assessed interventions combining education and contact content and obtained mixed effects. Staniland and Byrne (2013) evaluated an intervention to reduce stigmatising attitudes and behaviours towards autistic people. This covered education with both direct and video contact with autistic people. For attitudes, more positive attitudes were found in the intervention group and were maintained at the follow-up point ($p = .01$). As for behaviours, this intervention failed to work in the intervention group ($p = .22$), nor did it work when compared to the control group ($p = .37$). In addition, researchers examined the online activity usage effects and reported no pre-post differences between online activity users' attitudes and behaviours towards their autistic peers. DeLuca (2020) found that their intervention did significantly reduce anticipated stigma ($p = .020$) across test time points but not self-stigma. This intervention consisted of an educational presentation and a presentation of the personal story of the presenter.

Saporito et al. (2011) evaluated the efficacy of an intervention to decrease explicit and implicit stigma around adolescents' mental

health, consisting of a presentation with slides on mental health conditions in young people and a video presentation of a youth currently suffering from a mental health condition. The findings indicated that the intervention had effects on reducing explicit stigma (attitudes to mental health: $p = .03$; attitudes to treatment: $p = .001$) but not implicit stigma that was assessed by using automatic associations in memory related to help-seeking and people with mental health conditions. One study' intervention was education-based, and it had education plus contact as the comparator intervention (Chisholm et al., 2016). Those two were allocated to the education and contact condition received educational curriculum-based modules and a contact session working with a young person with experience of a mental health condition. The authors reported that their primary outcome, attitudinal stigma, was significantly decreased in both interventions (education-only condition or education plus contact intervention), but there was no significant effect of either intervention at a 2-week follow-up ($p = .5$). At a 6-month follow-up, a significant effect was shown in the education-only intervention compared with the education plus contact condition ($p = .03$).

One study had three interventions: education, contact and printed material (Painter et al., 2017). A curriculum with active learning and encouragement of empathy was delivered by teachers who introduced stigma-related knowledge and concepts of and coping with specific mental disorders. College students with histories of bipolar disorder were invited to do presentations in the contact section. Printed material consisted of posters focusing on individuals' personal traits and abilities as opposed to language that labels a person as 'mentally ill'. Finding showed that there was no effect from using posters ($p > .05$). Also, compared with the control group, the curriculum-only group had significantly more positive outcomes for 8 of 13 outcomes, and the contact-only group reported less effects than the curriculum-only group.

Two education-only interventions reported no effects. Townsend et al. (2019) implemented an intervention delivered by high school teachers and explored efficacy for increasing depression knowledge and reducing stigma. No main effect of the intervention on stigma scores ($p = .08$) were found. Howard et al. (2018) evaluated whether education information that described biological or psychological causes of mental-ill health could reduce stigma; neither information on biological nor psychological causes had significant effects on anticipated self-stigma or personal stigma. One study was a contact-related intervention that did not produce a positive effect. Pinto-Foltz et al.'s (2011) knowledge-contact programme involved narrative story, discussion, and video presentation. They found that stigma did not show reduction after the intervention ($p = .33$), and at 4- and 8-week follow-ups, there was no significant difference between adolescents in the intervention and control groups.

Intersectional analysis showed that four studies reported findings of gender difference in measured outcomes. Except for Townsend et al. (2019) who did not find a gender difference in stigma reduction, other studies consistently suggested that their interventions had more efficacy in females. Cheetham et al. (2020) reported that female stigma scores decreased over time, and females showed more positive attitudes towards stigma and less social distance than males in Economou et al.'s (2012) study. From baseline to follow-up, O'Mara et al. (2013) found that increased stigma scores were found in both males and females, but females increased less than males. No interventions reported details of any analysis or findings with regard to ethnic difference.

### Overview

According to these findings, education-based interventions were most likely to have positive effects on stigma reduction in young people, even though some education interventions showed mixed and no effect. Four studies (including Winkler et al., 2017) adopted contact interventions; two reported positive effects (Winkler et al., 2017; Mulfinger et al., 2018) and another two reported mixed (Vila-Badia et al., 2016) and no effect (Pinto-Foltz et al., 2011) respectively, making it hard to assess effectiveness of contact-only approaches for stigma reduction in this review. Education plus contact interventions could have positive effects on stigma reduction, but these were not significant or stable for long-term effects (e.g., Staniland and Byrne, 2013).

Effective intervention components were educational approaches, including lessons, curriculum that consisted of modules explaining stigma-related concepts and strategies, activities such as video games and facilitated discussion, which could be effective in reducing stigma through correcting misinformation on mental health. Also, contact with people with mental health conditions could potentially work for stigma reduction. Apart from one intervention, those with positive effects invited trained teachers or psychologists to deliver interventions, which may have contributed to the efficacy of these anti-stigma interventions.

### Meta-analysis

In total, 11 of the 22 studies did not report adequate statistics to be included in the formal analysis. Accordingly, a total of 11 studies were included in our meta-analytic models. Of these, six were education-only (Perry et al., 2014; Chisholm et al., 2016; Milin et al., 2016; Cangas et al., 2017; Howard et al., 2018; Nguyen et al., 2020), two were contact-only (Pinto-Foltz et al., 2011; Vila-Badia et al., 2016), two were education plus contact (Staniland and Byrne, 2013; DeLuca, 2020) and one (Winkler et al., 2017) had two anti-stigma interventions (seminar and short video) and an active control group (leaflets). Since some studies had more than one outcome, sample or intervention, these studies were further comprised of studies nested within them. For example, at *Post-Intervention*, Nguyen et al. (2020) reported results from a Vietnamese sample and a Cambodian sample, thus comprised two nested studies. In total, at *Post-Intervention,* there were 22 studies nested within the 11 parent studies selected, and at *Follow-Up,* there were 11 studies nested within the 6 parent studies.

At *Pre-Intervention*, the pooled effect size based on the three-level meta-analytic model was not significant ($d = .008$, $p = .856$). In other words, as expected, there was no discernible difference between control and intervention groups when tested *Pre-Intervention.* The multivariate model at *Post-Intervention* revealed a small, significant effect ($d = .21$, $p < .001$). Overall, heterogeneity was high ($Q(21) = 76.63$, $p < .001$). More specifically, the estimated variance components were $\tau^2_{\text{Level 3}} = 0.020$ and $\tau^2_{\text{Level 2}} = 0.017$, with $I^2_{\text{Level 3}} = 40.2\%$ of the total variation attributed to between-cluster, and $I^2_{\text{Level 2}} = 33.9\%$ to within-cluster heterogeneity. In other words, approximately a third of variance can be explained by differences *within* parent studies, with a slightly larger proportion of the variance accounted for by differences *between* parent studies. At *Follow-Up*, the multi-variate model revealed that interventions were no longer effective at reducing stigma ($d = .069$, $p = .347$). Funnel plots for both models were symmetrical (see Appendix 3 of the Supplementary Material), showing no evidence of publication bias, thus indicating no need for adjustment (Duval and Tweedie, 2000). Finally, pooled effect sizes for both *Post-Intervention* and

*Follow-Up* were considered in a model with time as a moderating factor. We found that the intercept (i.e., *Post-Intervention*) was significantly different from zero, with a small pooled effect size ($d = .212$, $p < .001$). Moreover, the coefficient (i.e., *Follow-Up*) was negative and significantly different from zero ($d = -.128$, $p = .046$), suggesting that there was a significant decline in the efficacy of intervention at *Follow-Up*. Once again, heterogeneity was high ($QE (31) = 99.37$, $p < .001$). More specifically, the estimated variance components were $\tau^2_{Level\ 3} = 0.021$ and $\tau^2_{Level\ 2} = 0.015$, with $I^2_{Level\ 3} = 41.26\%$ of the total variation attributed to between-cluster, and $I^2_{Level\ 2} = 29.3\%$ to within-cluster heterogeneity. Overall, the included interventions had a small effect reducing mental health stigma, but this decayed to no effect in the weeks following intervention delivery (see Figure 2).

### Risk assessment

Following the Cochrane guidance, the Risk of Bias 2 risk assessment tool was employed to assess risk of bias in randomised controlled trials and cluster-randomised trials (Sterne et al., 2019). The studies that were cluster-randomised trials design were assessed by Risk of Bias 2 for cluster-randomised trials and those randomised controlled trials and experiment designs with randomisation were evaluated by Risk of Bias 2. Overall, the included studies in this review indicated poor quality. No studies were rated in the low-risk category, few studies showed some concerns and others were evaluated as high risk of bias (see Appendix 2 of the Supplementary Material).

### Discussion

The study fills a gap in the current evidence on the effectiveness of anti-stigma interventions for young people and is the first to conduct a systematic review and meta-analysis exploring the global literature. Of the 22 studies included in this review, more than half of the interventions adopted education-based interventions. Review findings indicated that education-based interventions were most likely to have positive effects compared with contact-based or education plus contact interventions.

It has been argued that it is often ineffective to reduce stigma in the general public using educational programmes alone and that contact-based interventions are more successful than education-based interventions for adults (Corrigan et al., 2012). Our review showed the opposite findings for young people. A possible reason for this inconsistency is the target population: compared with adults, educational interventions might be more effective for youth because school is the place where young people easily have access to mental health knowledge, and it is routine for young people to have classroom-based learning (Mcluckie et al., 2014). There might also be cohort effects in school, where young people influence each other. Additionally, education-based interventions are relatively more economical and easier to deliver than contact-based interventions, for which it can be difficult and time-consuming to prepare people with mental health conditions to participate (Malachowski and Kirsh, 2013). This is evidence that young people could be suitable audiences for education-based interventions because these could help lay a solid foundation of having positive attitudes towards people with mental health conditions and prevent stigmatising behaviours in adulthood (Corrigan et al., 2005).

However, it is worth noting that six education-based interventions showed mixed effects and two were ineffective. This could be

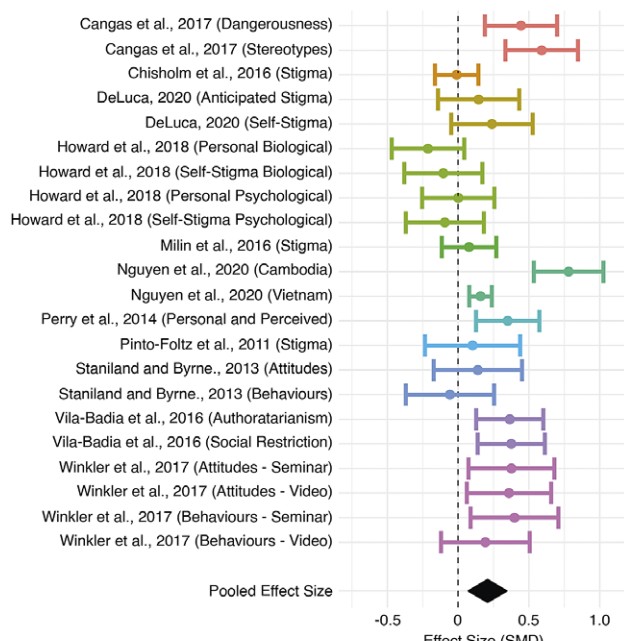

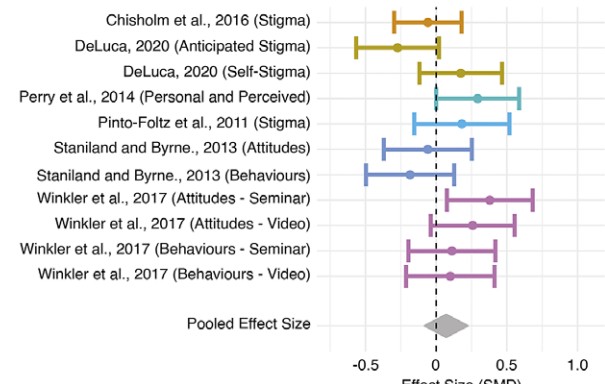

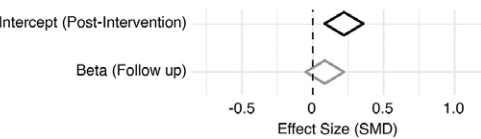

**Figure 2.** Effectiveness of anti-stigma interventions.
*Note*: Standardised mean differences are presented for each measure of stigma for each study at Post-Intervention (Top) and Follow-Up (Middle). Dots represent the overall effect size, with error bars showing 95% confidence intervals. The colours are common between studies nested within the same parent study. Pooled effect sizes calculated for each model (Separate: Top and Middle; As Moderating Effects: Bottom) are shown as diamonds, which are centred over the pooled effect size, extending as far as the 95% confidence interval.

related to intervention intensity and the role of trained facilitators. Some education-based interventions with mixed effects had low intensity with only one session (e.g., O'Mara et al., 2013). In education-based interventions with mixed/no effect(s), some were delivered by teachers who did not receive training (e.g., Painter et al., 2017), which could have affected intervention fidelity and efficacy. In contrast, trained teachers or psychology-related professionals participated in interventions with more efficacy (e.g.,

Winkler et al., 2017). Moreover, for interventions with positive effects, some interventions included several modules to improve the understanding of relevant concepts of mental health and coping strategies in youth. Students not only received basic knowledge on stigma from facilitators, but also engaged in facilitated discussions improving interactivity. These components that were interactively delivered could have contributed to efficacy (e.g., Milin et al., 2016). Although the previous findings showed the advantages of education-based interventions for stigma reduction by correcting stigmatising attitudes, there is no robust evidence supporting that specific non-stigmatising behaviours can be predicted from attitudes (Crano and Prislin, 2011). Thus, it is essential to explore other approaches to achieve destigmatising behaviours via further research.

Contact-based interventions did not demonstrate as much effectiveness as education-based interventions. The limited number of contact-based interventions that were included in this review could have made it difficult to evidence the same level of efficacy as for educational-based interventions. Included studies suggest that contact time could be a factor that affects efficacy (Gronholm et al., 2017), and longer contact time could be associated with more significant effects. For example, Mulfinger et al. (2018) had longer intervention contact time than Vila-Badia et al. (2016) did, and the former showed more positive effect than the latter. A possible explanation for the inconsistent findings in contact-based interventions could be the differences between measures for mental health stigma which assess different aspects of stigma. Although the included studies all aimed to evaluate interventions for reducing stigma, the measures used varied. For instance, Mulfinger et al. (2018) adopted the *Stigma Stress Scale*, Vila-Badia et al. (2016) used *The Community Attitudes towards Mental Illness* and Pinto-Foltz et al. (2011) employed a five-item subscale of the *Revised Attribution Questionnaire*. Also, interventions were not all delivered in a context with equivalent levels of stigma and it might be easier to show an effect when the levels of stigma are high at baseline. The education plus contact interventions showed mixed effects in this systematic review. This is inconsistent with previous studies showing that it is more effective to use a combination of education and contact interventions instead of employing each of these interventions alone (e.g., Chan et al., 2009). In Chan, Mak and Law's intervention, they had only one intervention group with three conditions but no control group, which could be a reason for the inconsistency.

Intersectional analysis within included studies focused only one gender, showing that interventions had more efficacy in females. Owing to a lack of detail on ethnicity and differences in stigma reduction within included studies, our review was not able to cover this aspect of diversity among young people. Further research is needed that includes more intersectional analysis of interventions for reducing stigma in young people, including any differences in interventions outcomes for people of different ethnicities.

Stigma reduction, it is argued, needs to be delivered in culturally specific ways, in order to align with 'what matters most' in given cultures, and the subsequent impact of stigma on people's ability to engage in 'what matters' in their society (Yang et al., 2007). With this in mind, the following three agendas (Corrigan, 2015; Corrigan and Al-Khouja, 2018) could be considered in future anti-stigma efforts to prevent young people from being excluded from 'what matters most' in society. A 'services agenda' is associated with promotion of mental health literacy and care-seeking; a 'rights agenda' achieves stigma reduction by affirming attitudes and behaviours to replace discrimination; a 'self-worth' agenda helps reduce self-stigma by promoting self-affirming attitudes in place of shame. Our reviewed findings indicate that interventions focused on these agendas included promoting mental health literacy, non-discriminatory attitudes towards mental health and reducing stigmatising behaviours, and reported positive effects in some studies (e.g., Milin et al., 2016; Cangas et al., 2017; Link et al., 2020). In particular, interventions based on a rights agenda and a self-worth agenda demonstrated more efficacy in this review. This suggests that such interventions would support young people with mental health conditions would be more able to take part in 'what matters most' in their culture. Intervention content would, however, need to be tailored to the form and causes of mental health stigma in any specific cultural context.

### Risk assessment

The overall quality of included studies was poor, and the primary concern was randomisation as only few of studies reported how allocation to intervention and control groups took place. Other concerns were 'Deviations from the intended interventions', which could be 'the administration of additional interventions that are inconsistent with the trial protocol, failure to implement the protocol interventions as intended or non-adherence by trial participants to their assigned intervention' (Higgins et al., 2016). In other assessment categories including report of missing data, measurement of outcome and results report, most studies were rated as having some concerns as there was a lack of information demonstrating how authors dealt with these issues. These omissions were possibly due to limited word counts for the publications, which adversely affected the risk assessment.

### Limitations

This is the first systematic review to look at effectiveness of anti-stigma interventions among young people. However, it has some limitations related to capacity within the research team. The grey literature was not searched, and aside from English and Chinese publications, studies written in other foreign languages were excluded for the review. This could have led to publication bias in terms of studies with null or negative results. The quality assessment was conducted by two assessors but may have been limited by only the first author being involved data extraction. This review did not differentiate between types of stigma, and further research could usefully investigate the effectiveness of anti-stigma interventions for different types of stigma. Moreover, the high heterogeneity means estimates of the pooled effect sizes from our models are less reliable than if study designs had been more similar. Finally, the overall poor quality of the included studies may mean that conclusions from the review are not generalisable and definitive recommendations on the effectiveness of anti-stigma interventions for young people require additional research.

### Implications

This review has provided evidence that anti-stigma interventions were effective overall but yielded small improvements that did not endure in the long term. Findings indicate that education-based interventions showed advantages for reducing stigma in youth compared with other interventions. Thus, incorporating education-related approaches, such as having interactive discussions, workshops and seminars, is recommended when developing anti-stigma interventions. Our results suggest the importance of

intervention intensity for education-based interventions. More sessions could contribute to positive and stable effects on reducing stigma in youth. Findings from the meta-analysis show that the intervention conducted by Cangas et al. (2017) had a larger effect than other approaches. This suggests that educational components could usefully include video games as an effective and innovative approach to reducing mental health stigma perpetuated by young people.

Schools have been identified as important sites to deliver mental health and well-being campaigns (Moore et al., 2022). Findings from this review confirm that schools are potentially effective sites from which to reach young people to reduce mental health stigma, however, the complexity of school settings needs to be considered. Prioritising other tasks, such as academic achievements, over mental health and well-being interventions, has been identified as an obstacle to effectiveness (Nadeem and Ringle, 2016; Dijkman et al., 2017; Crane et al., 2021). Staff turnover is another barrier to the sustainability of school-based interventions (Moore et al., 2022). Low intervention fidelity has been reported from school-based mental health and well-being interventions when an intervention was delivered partially by teachers who had not been trained or received materials (Friend et al., 2014). This is consistent with our review findings that those anti-stigma interventions that were delivered by trained teachers or psychology-related professionals were more effective. Additionally, school staff capacity to deliver interventions and the impact of this on intervention sustainability has been questioned (Moore et al., 2022). For example, in one study, adequate supervision was not provided by coordinators to teachers (Dijkman et al., 2017). There is evidence that engaged school leaders who were inclined to provide support and encouragement for school staff in the use of an intervention could facilitate its implementation (Hudson et al., 2020). This suggests that, to develop and deliver anti-stigma interventions in school settings, it is important to have support from school-based needs and decision-makers as a key factor.

The evidence base for anti-stigma interventions is itself a further issue highlighted by the review. With regard to the countries in which intervention studies were conducted in this review, only one study (Nguyen et al., 2020) was conducted in an Asian country and all other intervention studies were in Western countries. Thus, this review provides most evidence for anti-stigma interventions targeting youth in a Western context. To determine the components of effective anti-sigma interventions for youth in other contexts, and particularly in low- and middle-income countries, more studies on mental health stigma in young people need to be conducted in those settings.

**Open peer review.** To view the open peer review materials for this article, please visit http://doi.org/10.1017/gmh2023.34.

**Supplementary material.** The supplementary material for this article can be found at http://doi.org/10.1017/gmh2023.34.

**Data availability statement.** The data that support the findings of this study are available from the corresponding author, N.S., upon reasonable request.

**Acknowledgements.** This work was supported by Amy Palmer who was the second reviewer for screening full-text publications and Kier Harris who was the second assessor for a double quality assessment.

**Author contribution.** N. S. led the development of the protocol for this review, conducted the database searches, screened all studies, extracted data and drafted this article. G. M. contributed to the protocol development, double-screened 50% of included studies and contributed to the development of each draft of the manuscript. S. H.-J. contributed to the protocol development, double-screened

the remaining 50% of included studies and contributed to the development of each draft of the manuscript. J. P. conducted the statistical analysis and visualisation and contributed to the interpretation of the results. R. M. W. gave guidance on the statistical analysis and made final edits to the manuscript. All authors have approved the final article.

**Financial support.** This review did not receive any funding.

**Competing interest.** None of the authors has a competing interest related to the present study.

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
