## [Reviewer Report]

Dear authors, your article is of great scientific interest and relevance. The abstract and article are clearly written, giving a complete description and perspective.

However, the following comments and suggestions should be made explicit and incorporated:

1. Methods.

- You must explain and include the characteristics and specifications of the manual search performed.

- It is necessary to indicate the dates during which the search for articles was carried out, since this will provide information for the replicability of the study. Likewise, it is relevant to indicate the dates on which a manual search of the articles was carried out.

-Was a manual trace back literature search performed after the date of the search to identify new literature that may have been published?

-Since the search and selection of articles was carried out between 2 authors, it is relevant to indicate the degree of agreement between them, through the measurement of interrater reliability (e.g. using the Cohen κ coefficient).

- It is highly recommended to estimate the measure of publication bias (e.g. through the trim-and-fill method for publication bias).

2. Limitations:

- It is relevant to note that gray literature was not included in the search and selection of studies, which generates an increased risk of publication bias (because studies with null or negative results tend not to be published or submitted to scientific journals).

-It is not at all clear to me whether quasi-experimental studies (non-randomized but with control and experimental groups) were included, as this may bias the effect size and effectiveness of these studies.

---

## [Reviewer Report]

I appreciate the opportunity to review this timely and important paper on the effectiveness of anti-stigma interventions for reducing mental health stigma in young people. Thank you for allowing me to provide feedback and suggestions for improving the clarity and effectiveness of the paper.

Overall, the abstract provides a clear and well-written overview of the study, highlighting the issue of mental health stigma among young people and the effectiveness of anti-stigma interventions. The introduction effectively sets the stage for the paper, citing relevant studies and research to support the prevalence of mental health disorders and the impact of stigma on seeking help and receiving treatment.

The methods section clearly explain how the paper follows established guidelines for systematic reviews, and the results section provides sufficient detail on the interventions and their effectiveness in reducing mental health stigma.

Some things to consider:

Some concepts introduced in the results section may be confusing and require further explanation. Some examples: “Stigma stress” (l 334), “benevolence” (l 366), “community mental health ideology” (l 366), etc. These concepts come from the reviewed articles but should be explained to ensure everything is clear.

In the discussion, lines 559-562, it is indicated that it is more challenging to reduce stigma in the USA and Australia compared to other countries in the review. This claim should be substantiated.

The study found that education-based interventions were most likely to have positive effects compared to contact-based or education plus contact interventions. It is argued that this is due to the target population, young people, may be more receptive to education-based interventions because school is the place where they can easily access mental health knowledge. The discussion section could benefit from further elaboration on the implications of working more closely with schools for awareness raising in stigma. The complexity of schools as over-interveened spaces with different normative agendas, and with staff working in precarious conditions should also be recognized.

The existence of stigmatizing thoughts and attitudes towards persons with mental illness responds to each context’s cultural and social features. Link’s work around “what matters most” in different contexts points to this fundamental fact. It reveals that mental illness does not necessarily mean stigma: what is stigmatized depends on what is valuable about being a human in different settings and what is expected from fellow humans. This also means that public action against stigma has different forms and is moved by different agendas depending on each context. Sheehan and Corrigan have identified a) A services agenda, which seeks to decrease stigma so people better engage in care, b) a rights agenda, which seeks to stem the injustice of stigma so people are able to meet life goals and aspirations, and c) self-worth agenda, which strives to replace shame with self-affirming attitudes in people impacted by stigma.

I ask the authors to engage with these aspects of stigma, recognizing that what stigma is, how it should be addressed, and the effectiveness of different strategies in the long and short run is all context-dependent.

Overall, the paper is well-written and informative, but could benefit from further elaboration and explanation on some concepts and implications.

---

## [Reviewer Report]

Dear authors,

Thank you for submitting this manuscript.

It addresses a relevant issue in the field as there is a need to learn more about what works to reduce stigma in diverse contexts.

The reviewers have noted, however, that your manuscript could be benefit from further elaboration and clarification of key concepts, methods, and findings.

Looking forward to reading the revised manuscript.

Best,

Franco Mascayano

---

## [Reviewer Report]

Dear authors, I appreciate the effort made and the inclusion of the reviewers' comments in your article.

---

## [Reviewer Report]

I am pleased to say that the authors have thoroughly addressed the concerns raised in my initial review. The revisions and amendments made have significantly improved the overall quality of the manuscript. Consequently, I believe it is now in a suitable state for publication.

This piece does an excellent job synthesizing a wide array of research in an accessible manner, making it a significant contribution to our current understanding of school interventions for reducing stigma. The manuscript’s focus on young people’s mental health stigma - a topic of undeniable importance - further enhances its value.

I would like to express my gratitude for the opportunity to review this piece. It has been a privilege to be a part of the process and to witness the development of such a critical piece of scholarship.

Thank you once again for the opportunity. I look forward to seeing this work shared with the wider academic community.